# Position: Your VLM May Not Be Thinking with Interleaved Images

**Wenjie Yang** [* 1]  **Siqi Zhu** [* 1]  **Zengfeng Huang** [1 2]

## Abstract

"Thinking with Images" has emerged as a central research theme in the realm of Vision-Language Models (VLMs). This multimodal reasoning paradigm typically features interleaved images generated via tool use or code execution as part of the Chain-of-Thought (CoT). While reinforcement learning (RL) has driven impressive performance within this paradigm, **in this position paper, we argue that current VLMs seldom truly "think" with interleaved images.** Through empirical evidence and analysis, we demonstrate that interleaved images do not play a significant role in the success of recent "Thinking with Images" methods. Instead, the primary source of performance gains is the improved language generation distribution resulting from finetuning. These findings challenge the prevailing belief that "Thinking with Images" VLMs actively utilize visual information to complete visual tasks. To improve mechanistic transparency, we suggest that future "Thinking with Images" works include lightweight ablation studies to verify the necessity of interleaved images. Furthermore, we call upon the community to develop fundamentally novel benchmarks that truly require visual reasoning, and advocate for more informative visual tools.

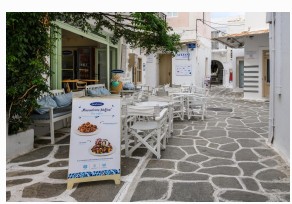

Input Image

**Query:** What is the color of the dress?

**Thought:** *The image shows a narrow street with white-washed buildings, outdoor seating areas, and a signboard advertising mini waffles. The scene appears to be part of a quaint town or village setting, possibly in Greece due to the architectural style and signage. Now I will zoom in to look clearer at the dress.*

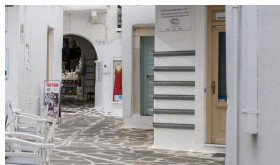

Interleaved Image

**Answer:** The cropped image focuses on the right side of the street, where a door is visible. Above the door, there is a display window showing a red dress hanging inside.

*Figure 1.* **Illustration of the "Thinking with Images" framework.** Ideally, "Thinking with Images" models leverage tools or code to generate interleaved images that facilitate visual reasoning tasks. This example demonstrates a successful reasoning trajectory where the model employs a zooming tool to resolve the query.

## 1. Introduction

VLMs have achieved substantial progress in recent years (Liu et al., 2023; Bai et al., 2025b;a; Wang et al., 2025d). A particularly prominent milestone is OpenAI o3 (OpenAI, 2025), which pioneered research into "Thinking with Images". "Thinking with Images" represents a new paradigm for solving visual tasks by leveraging visual information as

an integral part of the CoT. Figure 1 illustrates an example of "Thinking with Images": when queried about the color of a small target in a dense, high-resolution image, the VLM utilizes a zooming tool to acquire a crop containing the target before providing the correct answer. Such compelling demonstrations have prompted many researchers to explore this field (Zhu et al., 2025; Zhang et al., 2026; Su et al., 2025a; Hou et al., 2025; Guo et al., 2025). Consequently, numerous attempts have been made to apply this paradigm across various domains, including GUI agents (Lei et al., 2025), grounding (Yang & Huang, 2025), and geolocalization (Ji et al., 2026). Most of these works employ tools or code to generate interleaved images, inserting them into the CoT as intermediate visual information. However, maintain-

---

[*]Equal contribution, alphabetical order [1]School of Data Science, Fudan University, Shanghai, China [2]Shanghai Innovation Institute, Shanghai, China. Correspondence to: Wenjie Yang <yangago668@gmail.com>, Zengfeng Huang <huangzf@fudan.edu.cn>.

*Proceedings of the 43rd International Conference on Machine Learning*, Seoul, South Korea. PMLR 306, 2026. Copyright 2026 by the author(s).

ing such interleaved images is computationally expensive, as they significantly increase the context length for VLMs. Furthermore, they dramatically increase the difficulty of finetuning, as VLMs require specialized training to master appropriate tool-calling and code generation strategies (Zhang et al., 2026; Zheng et al., 2026; Su et al., 2025a).

Given the impressive performance reported on various benchmarks (e.g., V*(Wu & Xie, 2024) and HR-Bench (Wang et al., 2025c)), there is a common belief that the interleaved images in these "Thinking with Images" CoTs are, at least to some extent, essential. However, **this position paper argues that "Thinking with Images" VLMs seldom truly "think" with these interleaved images.** We support this argument through both empirical evidence from multiple perspectives and reasoning grounded in comprehensive literature. By taking this position, we aim to spark more insightful discussions within the "Thinking with Images" field, particularly regarding the genuine utility of interleaved images and the direction of future research. We particularly emphasize that there is an urgent need for novel benchmarks that genuinely require visual reasoning. To facilitate these discussion, we structure the paper as follows.

**Outline.** In Section 2, we review recent developments in "Thinking with Images" and discuss related works that is relevant to our topic. Section 3 provides a formal definition of "Thinking with Images". In Section 4, we present two key empirical findings, namely, the ablation of interleaved images and attention visualization, to demonstrate that interleaved images do not play a major role in current "Thinking with Images" methods. Section 5 investigates the underlying causes of the success reported by recent "Thinking with Images" works. This section also includes a bonus subsection outlining a straightforward method for enhancing benchmark performance. Finally, we offer actionable suggestions in Section 6 and address potential counterarguments in Section 7.

## 2. Related Work

**"Thinking with Images" methods.** Recent studies on VLMs have achieved impressive performance across various visual tasks by employing multimodal CoT (Zheng et al., 2026; Zhang et al., 2026; Su et al., 2025a). These models, which reason using interleaved text and images, are termed "Thinking with Images" (Su et al., 2025b) methods. Unlike previous paradigms where visual information serves merely as static input, "Thinking with Images" methods introduce a novel approach to actively perceive and manipulate images. A prominent line of work utilizes active perception, where the model learns to dynamically control its visual focus via cropping and zooming. For instance, Active-o3 (Zhu et al., 2025) applies a cropping tool driven by a sensing policy to propose target regions. Similarly,

DeepEyes (Zheng et al., 2026) employs reinforcement learning to intrinsically incentivize the grounding ability of the model, while Pixel-Reasoner (Su et al., 2025a) introduces a curiosity-driven RL method to encourage tool usage. Another line of research leverages code execution for image manipulation, enabling models to develop new tools when appropriate. Among these, Thyme (Zhang et al., 2026) achieves promising results with a standard yet delicate supervised finetuning (SFT)-then-RL paradigm. Building on this, subsequent works have explored more diverse tools. For example, CodeVision (Guo et al., 2025) addresses rotation for non-level images, VTool-R1 (Wu et al., 2026) uses the highlight tool, and VACoT (Xu et al., 2025) introduces denoising with a focus on OCR-related tasks. While these image manipulations appear basic, training VLMs to utilize them accurate remains a significant challenge. Moreover, incorporating interleaved images imposes substantial computational overheads during both training and inference. Given the impressive results achieved by pioneering works in "Thinking with Images", a natural hypothesis is that these costly interleaved images are instrumental to reasoning (e.g., allowing the VLM to "zoom in" for better visibility). However, we demonstrate that, contrary to this belief, interleaved images do not play a major role in most current use cases.

As "Thinking with Images" is a rapidly expanding field, an exhaustive evaluation of every method is infeasible. In this paper, we specifically focus on prestigious reinforcement learning-based works that represent the state-of-the-art, while also acknowledging other paradigms such as "Thinking with Images" using image generation (Gu et al., 2026; Wen et al., 2026) and latent "Thinking with Images" (Wang et al., 2025b).

**Rethinking (visual) reasoning models.** There is a growing body of work rethinking the necessity of long CoT for LLMs and VLMs. In the pure text domain, several studies indicate that longer CoT does not always yield better results (Hassid et al., 2025; Wu et al., 2025b). This discussion has led to adaptive CoT approaches that balance effectiveness and efficiency (Yu et al., 2025; Sun et al., 2025). In the context of latent reasoning, research suggests that latent tokens may not always encode faithful reasoning (Zhang et al., 2025c). Within the "Thinking with Images" domain, insightful works have also questioned the necessity of lengthy CoT. On maze-solving benchmarks, Du et al. (2025) show that while visual and longer CoT accelerate convergence, they do not necessarily boost performance. Through causal intervention, Liu et al. (2025c) demonstrate that visual information in CoT has less impact than textual reasoning. Furthermore, Wang et al. (2025a) and Hou et al. (2025) report that some visual information in "Thinking with Images" methods is unfaithful: for instance, image crops generated by tools may not contain the target objective. These findings provide crucial insights for piecing together a comprehensive view.

And there are also several concurrent works that also investigate the effectiveness of zooming and cropping (Ma et al., 2026; Wei et al., 2026; Song et al., 2026). In this work, we systematically investigate the role of interleaved images with extensive empirical evidence and provide actionable suggestions for future research.

**Analysis on visual tools.** Discussing visual tools in the context of "Thinking with Images" is essential, as most frameworks obtain interleaved images via tool or code execution. Some works studying cropping and zooming, the most common tools in "Thinking with Images" methods, are particularly relevant. Zhang et al. (2025a) shows VLMs actually "know where to look" even when they fail to answer correctly, a training-free method called Vicrop is proposed based on this observation. Following Vicrop, HiDe (Liu et al., 2025a) decouples zooming into upscaling and cropping, showing that while scaling images offers no performance gain, separating the target objective from the background is highly beneficial. Additionally, Liu et al. (2025b) show that perceiving the objective is generally not difficult for VLMs, and failures on visual tasks often arise from ineffective leverage of visual information. Taken together, we can understand from first principles why zooming may be ineffective for perception tasks: image scale is not critical because popular VLMs like Qwen2.5-VL (Bai et al., 2025b) process images as patches, and training data is diverse enough to cover various scales. Essentially, perception tasks are challenging because they function as long-context problems for the language model, where image patches containing the target objective represent relevant context, while textual tokens and other patches act as noise. While finetuning improves the robustness of "Thinking with Images" methods to noisy context, cropping does not necessarily improve context management, as it introduces additional textual tokens while merely duplicating useful patches.

## 3. Preliminaries

We follow the common framework of "Thinking with Images" throughout this paper (Su et al., 2025b). Given an input query $Q$ and input images $I$, a VLM $\pi_\theta$ is required to generate a sequence of reasoning steps conditioned on the previous history $S_t = (z_1, \ldots, z_{t-1})$. Let $\mathcal{T}$ denote the space of all possible textual outputs and $\mathcal{I}$ denote the space of all possible interleaved visual outputs. The next step $z_t$ is generated according to

$$z_t \sim P(z_t \mid S_t, I, Q; \pi_\theta), \tag{1}$$

where $z_t \in \mathcal{T} \cup \mathcal{I}$. We refer to the subset $\{z_t | z_t \in \mathcal{I}\}$ as *interleaved images*. This framework covers a spectrum of "Thinking with Images" methods, including DeepEyes (Zheng et al., 2026), in which interleaved images are generated via external tools, and Thyme (Zhang et al., 2026),

where interleaved images are produced through code execution.

Throughout this paper, we primarily focus on the reinforcement learning based implementations of "Thinking with Images", e.g., DeepEyes (Zheng et al., 2026), Pixel-Reasoner (Su et al., 2025a), and Thyme (Zhang et al., 2026), as this paradigm currently represents the state-of-the-art performance in the field. These models are finetuned from the open-source VLM Qwen2.5-VL-7B-Instruct (Bai et al., 2025b) using Group Relative Policy Optimization (GRPO) (Shao et al., 2024). For each input $(I, Q)$, GRPO samples $G$ rollouts $\mathbf{S} = \{S_1, S_2, \cdots, S_G\}$ from the old policy $\pi_{\text{old}}$ and optimizes the model by maximizing the following objective:

$$\mathcal{L}_{\text{GRPO}}(\theta) = \mathbb{E}\left[\frac{1}{G}\sum_{i=1}^{G}\frac{1}{|S_i|}\sum_{t=1}^{|S_i|}\left\{\min\left[r_t(\theta)\hat{A}_{i,t},\right.\right.\right.$$
$$\left.\left.\left.\text{clip}\big(r_t(\theta), \epsilon\big)\hat{A}_{i,t}\right] - \beta\mathbb{D}_{\text{KL}}[\pi_\theta||\pi_{\text{ref}}]\right\}\right], \tag{2}$$

where $r_t(\theta) = \frac{\pi_\theta(S_{i,t}|(I,Q), S_{i,<t})}{\pi_{\text{old}}(S_{i,t}|(I,Q), S_{i,<t})}$ is the ratio function, $\hat{A}_{i,t} = \frac{r_i - \text{mean}(\mathbf{r})}{\text{std}(\mathbf{r})}$ is the advantage that is computed with the rewards $\mathbf{r} = \{r_1, r_2, \cdots, r_G\}$, $\epsilon$ and $\beta$ are hyperparameters, $\pi_{\text{ref}}$ is the reference model, and $\mathbb{D}_{\text{KL}}$ is the unbiased estimator of the KL divergence.

## 4. The Necessity of Interleaved Images

The interleaved images in the CoTs of "Thinking with Images" methods are costly to support. During training, they complicate data curation for SFT and require special agentic loops and rewards in RL. At inference, the increased context length adds further computational cost. Therefore, it is critical to explore whether these interleaved images are truly necessary in the "Thinking with Images" paradigm. In this section, we investigate this from two perspectives: (i) we test whether removing these images causes significant performance drops on popular benchmarks, and (ii) we visualize attention heatmaps on both input and interleaved images to provide an intuitive understanding of where "Thinking with Images" VLMs focus their attention.

### 4.1. Ablation of Interleaved Images

We begin with an ablation study on interleaved images to assess their importance in the CoT of "Thinking with Images" methods. Specifically, we suppress the image outputs produced by external tools or code executions, while retaining the preceding textual reasoning process. This design allows us to isolate and quantify the contribution of visual information itself. Such experiments provide a strong indication of the utility of interleaved images and are commonly

*Table 1.* **Ablation study of interleaved images within the visual CoT of "Thinking with Images" models.** All results are obtained using VLMEvalKit (Duan et al., 2024). The values in parentheses indicate the performance difference compared to the full model.

| Method | V* | HRbench-4K | HRbench-8K | MME-Real-Lite |
|---|---|---|---|---|
| Qwen2.5-VL-7B | 76.4 | 68.1 | 65.5 | 44.5 |
| DeepEyes-7B | 84.3 | 72.8 | 69.3 | 53.9 |
| w/o Interleaved images | 84.3 (0.0) | 72.3 (-0.5) | 69.5 (+0.2) | 53.5 (-0.4) |
| Pixel-Reasoner-7B | 85.3 | 72.5 | 68.9 | 50.0 |
| w/o Interleaved images | 83.8 (-1.5) | 72.5 (0.0) | 67.9 (-1.0) | 49.3 (-0.7) |
| Thyme-7B | 83.8 | 78.3 | 72.3 | 53.8 |
| w/o Interleaved images | 83.2 (-0.6) | 78.5 (+0.2) | 72.4 (+0.1) | 53.9 (+0.1) |

| Method | ChartQA | OCRBench | LogicVista | MathVision |
|---|---|---|---|---|
| Qwen2.5-VL-7B | 86.1 | 88.2 | 46.1 | 26.6 |
| DeepEyes-7B | 86.1 | 85.3 | 44.7 | 26.0 |
| w/o Interleaved images | 86.1 (0.0) | 85.0 (-0.3) | 43.4 (-1.3) | 26.6 (+0.6) |
| Pixel-Reasoner-7B | 87.1 | 82.1 | 42.1 | 27.6 |
| w/o Interleaved images | 87.1 (0.0) | 82.5 (+0.4) | 41.6 (-0.5) | 27.3 (-0.3) |
| Thyme-7B | 87.7 | 86.7 | 50.6 | 25.7 |
| w/o Interleaved images | 87.7 (0.0) | 86.8 (+0.1) | 50.3 (-0.3) | 25.7 (0.0) |

regarded as ground truth for explainability in prior work (Abnar & Zuidema, 2020; Chefer et al., 2021a;b). We use the following benchmarks: V* (Wu & Xie, 2024), HR-Bench 4K&8K(Wang et al., 2025c), MME-Realworld-Lite (Zhang et al., 2025d), ChartQA (Masry et al., 2022), OCRBench (Liu et al., 2024), LogicVista (Xiao et al., 2024), and MathVision (Wang et al., 2024). Table 1 presents the results for three recent RL-based "Thinking with Images" methods: DeepEyes-7B (Zheng et al., 2026), Pixel-Reasoner-7B (Su et al., 2025a), and Thyme-7B (Zhang et al., 2026). We also include the performance of Qwen2.5-VL-7B (Bai et al., 2025b), which serves as the starting checkpoint for training these three models. **We observe that removing interleaved images does not lead to significant degradation. In fact, it occasionally yields comparable or even superior accuracy.** On perception benchmarks, the slight performance difference caused by this ablation is negligible when compared to the substantial gains achieved through finetuning over the original Qwen2.5-VL-7B. This suggests that the success of "Thinking with Images" may not primarily stem from the interleaved images generated via cumbersome tool execution. It is worth noting that Liu et al. (2025c) provide relevant insights into causality, demonstrating that perturbing interleaved images causes significantly less performance degradation than perturbing textual reasoning. We take a more radical step by entirely removing these images and assessing the effect on tasks beyond the perception domain.

**Qwen3-VL-235B-A22B-Instruct** (Bai et al., 2025a) is a recent, popular VLM with "Thinking with Images" capabil-

*Table 2.* **Results on HRBench (Wang et al., 2025c).** Qwen3-VL denotes the Qwen/Qwen3-VL-235B-A22B-Instruct checkpoint. $\Delta$ denotes the performance difference. The raw performance is reproduced by us, while the "Thinking with Images" result is reported by the official technical report of Qwen3-VL. We only report HRBench since the reproduced results of other benchmarks are not available.

| Method | HR-4K | HR-8K | Overall |
|---|---|---|---|
| Qwen3-VL (Raw) | 85.6 | 82.3 | 84.0 |
| +Think w image | 85.4 | 82.4 | 83.9 |
| $\Delta$ | +0.2 | -0.1 | +0.1 |

ities, making it of particular interest to investigate whether our observations hold for this model. We evaluate it on the HR-Bench4K&8K(Wang et al., 2025c) since the setting on this benchmark is clear [1]. The results are presented in Table 2. We use Lmms-eval (Zhang et al., 2024) as the evaluation framework as suggested by the Qwen3VL team. The result shows that Qwen3-VL-235B-A22B-Instruct performs similarly, if not better, on perception benchmarks, which aligns with our previous findings for "Thinking with Images"-only VLMs. However, we would like to emphasize that this does not imply that training "Thinking with Images" models is without merit. The goal of this paper is to highlight that performance gains may not necessarily stem from the interleaving of images and text. Further experimentation

---

[1]https://github.com/QwenLM/Qwen3-VL/issues/1719

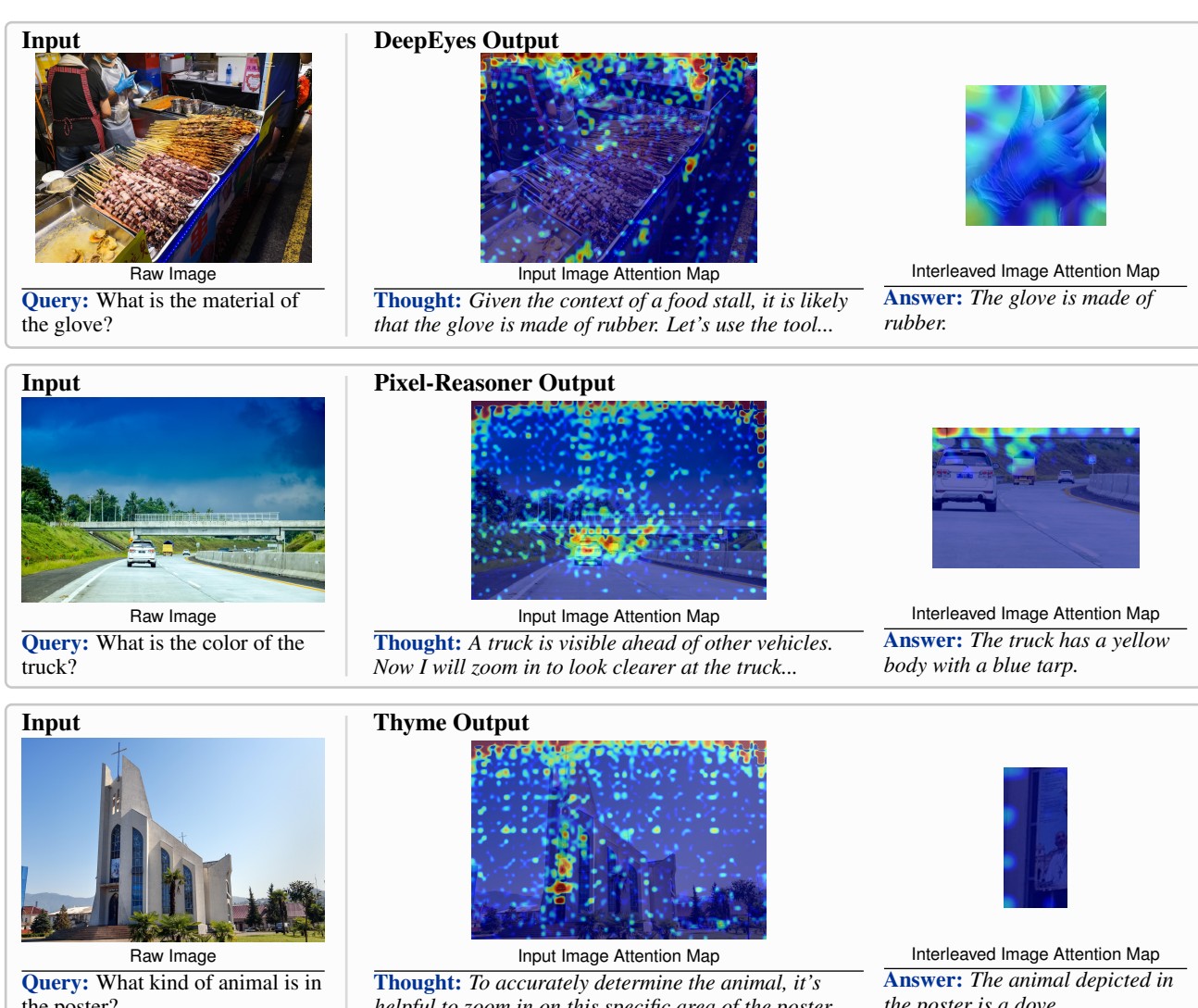

*Figure 2.* **Attention maps of "Thinking with Images" models.** We visualize the attention heatmaps on both the input and interleaved images, indicating the spatial focus of the answer tokens. These scores are computed using attention rollout (Abnar & Zuidema, 2020). Note that the generation sequence proceeds as follows: **Thought** text, followed by the interleaved image, and finally the **Answer** text.

is needed to better understand the optimal ways to utilize visual information in these models.

## 4.2. Attention Visualization

Another interesting analysis involves attention visualization (Abnar & Zuidema, 2020; Chefer et al., 2021a;b), which provides intuitive insights into where the VLMs actually "see". To better quantify the information flow, we utilize the attention rollout technique (Abnar & Zuidema, 2020). Given the attention matrix $\mathbf{W}(l_i)$ at layer $l_i$, the attention rollout is computed as follows:

$$\tilde{\mathbf{A}}(l_i) = \mathbf{A}(l_i)\mathbf{A}(l_{i-1})\ldots\mathbf{A}(l_0), \qquad (3)$$

where $\mathbf{A}(l_i)$ is the raw attention with a residual connection, i.e., $\mathbf{A}(l_i) = 0.5\mathbf{W}(l_i) + 0.5\mathbf{I}$ and $\mathbf{I}$ is the identity matrix. The derivation of attention rollout is intuitive when the attention mechanism is viewed as propagation over a weighted graph. In such case, the power of adjacency matrix, i.e., the attention matrix, represents the connectivity between nodes, i.e., tokens. We refer to the original paper (Abnar & Zuidema, 2020) for further details. Figure 2 presents some examples of attention visualization applied to the interleaved visual CoT produced by DeepEyes (Zheng et al., 2026), Pixel-Reasoner (Su et al., 2025a), and Thyme (Zhang et al., 2026). **We observe that these models predominantly attend to the original input images rather than the interleaved images.** While attention visualizations provide

qualitative insights, we consider the quantitative results from the ablation study to be more robust evidence of the model behavior. There are also other visualization techniques, but exploring them is not the main goal of this paper.

Another notable observation lies in the reasoning process of DeepEyes: the model arrives at the correct answer before utilizing the zoom-in tool. It first identifies the scene as a food stall and then leverages textual priors to infer the material. Contrary to previous studies that attribute hallucinations to textual priors (Zhang et al., 2025b; Deng et al., 2025; Liu et al., 2025b), this example demonstrates the beneficial role of such reasoning patterns. We will demonstrate later that this factor is crucial for understanding the performance gains of "Thinking with Images" methods.

> **Takeaway 1**
>
> Interleaved images are not the primary factor in the success of recent "Thinking with Images" VLMs on current benchmarks.

## 5. The Underlying Reason of Performance Gain

After demonstrating that interleaved images are not the primary factor driving the success of recent "Thinking with Images" models, a natural question arises: *what is the true source of the performance gains achieved by these methods?* We note that most recent "Thinking with Images" models, e.g., DeepEyes (Zheng et al., 2026), Pixel-Reasoner (Su et al., 2025a), and Thyme (Zhang et al., 2026), are finetuned on high-quality datasets that closely resemble the evaluation benchmarks. Therefore, it is possible that the finetuning itself brings the superior performance. To test this hypothesis, we run simple finetuning experiments that exclude any CoT annotations containing interleaved images.

### 5.1. The Effectiveness of Vanilla SFT

We apply the most straightforward form of supervised finetuning, which minimizes the sentence-level cross-entropy:

$$\mathcal{L}_{\text{SFT}}(\theta) = \mathbb{E}_{(I,Q,S_T)\sim\mathcal{D}}[-\log \pi_\theta(S_T|I,Q)], \quad (4)$$

where $\mathcal{D} = \{(I, Q, S_T)\}$ denotes the finetuning data. Crucially, the target sequence $S_T$ represents a concise natural language response that directly addresses the query (e.g., a brief factual description or a direct answer sentence), rather than a mere classification label. We directly adopt the DeepEyes data (Zheng et al., 2026), which is originally used for training the DeepEyes-7B models. Notably, this dataset contains neither interleaved images nor textual CoT annotations, as it was originally constructed for reinforcement learning.

*Table 3.* **Effectiveness of vanilla SFT.** DeepEyes-7B is the "Thinking with Images" model trained by Zheng et al. (2026) that depends on long CoT with tool-generated interleaved images. DeepEyes-SFT-7B is trained by us using vanilla SFT, without reasoning traces or interleaved images. Both models are finetuned from the same backbone, Qwen2.5-VL-7B (Bai et al., 2025b). $\Delta$ reports the performance difference between DeepEyes-SFT-7B and DeepEyes-7B. Avg. denotes the macro average of the 8 benchmarks. The performance gap on MME-RealWorld-Lite (Zhang et al., 2025d) likely stems from its stringent reasoning requirements, for which RL proves more effective than SFT.

| Method | V* | HR-4K | HR-8K |
|---|---|---|---|
| Qwen2.5-VL-7B | 76.4 | 68.1 | 65.5 |
| DeepEyes-7B | 84.3 | 72.8 | 69.3 |
| DeepEyes-SFT-7B | 85.3 | 72.9 | 70.8 |
| $\Delta$ | +1.0 | +0.1 | +1.5 |
| Method | MME | ChartQA | OCRBench |
| Qwen2.5-VL-7B | 44.5 | 86.1 | 88.2 |
| DeepEyes-7B | 53.9 | 86.1 | 85.3 |
| DeepEyes-SFT-7B | 49.1 | 86.5 | 87.4 |
| $\Delta$ | -4.8 | +0.4 | +2.1 |
| Method | LogicV | MathV | Avg. |
| Qwen2.5-VL-7B | 46.1 | 26.6 | 62.7 |
| DeepEyes-7B | 44.7 | 26.0 | 65.3 |
| DeepEyes-SFT-7B | 46.8 | 27.3 | 65.8 |
| $\Delta$ | +2.1 | +1.3 | +0.5 |

We use this data to finetune Qwen2.5-VL-7B for one epoch. The results are reported in Table 3. **We observe that vanilla SFT achieves performance comparable to, and in some cases higher than, its "Thinking with Images" counterpart.** Combined with our earlier findings that interleaved images contribute little to the overall performance gains, these results suggest that the observed improvements may be largely attributable to the finetuning process itself, rather than to the use of interleaved images or complex reasoning traces. We also emphasize that we deliberately choose SFT over reinforcement learning, as our goal is to avoid encouraging models to produce long CoT reasoning, which makes the analysis more controlled and interpretable. At the same time, we acknowledge a recent impressive work by Du et al. (2025), which achieves comparable performance using textual reinforcement learning.

**Implementation Details.** We employ LlamaFactory (Zheng et al., 2024) to finetune the Qwen2.5-VL-7B model (Bai et al., 2025b). The training utilizes a cosine learning rate scheduler with a peak learning rate of 1e-5. We set the per-device batch size to 16, resulting in a global batch size of

*Table 4.* **Results on V\* (Wu & Xie, 2024) with masked images.** We mask the regions containing the target object in each image. "Attribute" denotes the attribute recognition task, and "Spatial" denotes the spatial relationship task.

| Method | Attribute | Spatial | Overall |
|---|---|---|---|
| Qwen2.5-VL-7B | 14.8 | 50.0 | 28.8 |
| DeepEyes-7B | 23.5 | 59.2 | 37.7 |
| Pixel-Reasoner-7B | 26.1 | 55.3 | 37.7 |
| Thyme-7B | 31.3 | 55.3 | 40.8 |

128 across 8 GPUs. The model is finetuned for one epoch, and the training process completes within 15 minutes on a server equipped with H200 GPUs.

> **Takeaway 2**
>
> On current benchmarks, SFT can yield performance comparable to that of "Thinking with Images" models, indicating that these benchmarks may not be the optimal testbed for visual reasoning.

### 5.2. The Masked Experiment

We have shown that vanilla SFT can achieve performance comparable to that of "Thinking with Images" models. However, one might argue that vanilla SFT and "Thinking with Images" methods achieve similar performance through different mechanisms. To address this, we take a more radical step by conducting a masked experiment. Specifically, for all images in the V\* benchmark (Wu & Xie, 2024), we manually annotate bounding boxes corresponding to the target objects. These regions are then masked to occlude the visual targets, thereby forcing the VLMs to rely solely on the background context and infer the answers based on their internal textual priors.

We test the three "Thinking with Images" VLMs used in Section 4: DeepEyes-7B (Zheng et al., 2026), Pixel-Reasoner-7B (Su et al., 2025a), and Thyme-7B (Zhang et al., 2026). Since all three models are fine-tuned from Qwen2.5-VL-7B (Bai et al., 2025b) using reinforcement learning, we also report the performance of the base Qwen2.5-VL-7B model. **We observe that recent "Thinking with Images" models outperform Qwen2.5-VL-7B even when the target objects are not visible in the input images.** In this setting, the active perception capability of "Thinking with Images" methods is rendered ineffective. Thus, it is safe to conclude that finetuning aligns the language generation distribution better for this benchmark.

> **Takeaway 3**
>
> Finetuning provides VLMs with a language generation distribution that better aligns with the benchmark.

## 6. Call to Action

The aim of this paper is to assist the community in building efficient multimodal models that accomplish vision tasks without redundant effort. To realize this goal, we propose the following lightweight actions. These recommendations are designed to benefit the development of the field while imposing minimal burden on researchers.

- **"Thinking with Images" papers should report ablation studies on interleaved images.** Previously, "Thinking with Images" papers (Zheng et al., 2026; Su et al., 2025a; Zhang et al., 2026) have typically reported ablation studies only on the training algorithm (e.g., the performance of text-only reinforcement learning). However, such studies are sensitive to training setups, and the necessity of interleaved images cannot be fully justified by these results alone. In contrast, ablation studies on interleaved images, as shown in Table 1, are easy to implement, as one need only suppress the output of the tool/code. This approach is also cost-effective, requiring only inference adjustments. The genuine utility of interleaved images is validated only when their removal leads to a consistent drop in performance.

- **"Thinking with Images" methods should utilize tools that provide novel information.** Current "Thinking with Images" approaches primarily employ cropping as a tool to mimic human zoom-in behaviors (Zhang et al., 2026; Su et al., 2025a; Zheng et al., 2026). However, cropping may be ineffective for VLMs, as their visual encoders do not process information in a humanoid manner (Liu et al., 2025b;a). Consequently, the limited impact of interleaved images in current research may be attributed to the limited information provided by the tool itself. Following practices in textual tool-use (Schick et al., 2023; Qin et al., 2024), we recommend integrating tools that introduce new information, such as online search (Wu et al., 2025a; Hong et al., 2026) and image editing or generation (Gu et al., 2026; Tong et al., 2025).

- **"Thinking with Images" domain requires challenging benchmarks that necessitate visual thinking.** Benchmarks play a pivotal role in guiding the development of a field. Currently, "Thinking with Images" models are primarily evaluated on VQA tasks that do not necessarily require complex visual reasoning.

While there have been admirable attempts to construct more challenging benchmarks, such as RealX-Bench (Hong et al., 2026), merely combining multiple difficulties within a single sample does not fundamentally alter the nature of the task. We recommend adopting tasks that strictly require visual feedback as a testbed for "Thinking with Images" methods, such as irregular maze solving and geometric editing (Wei et al., 2025). There are two very recent benchmarks that we want to highlight, namely BabyVision (Chen et al., 2026) and InkSlop (Maksai, 2026). BabyVision evaluates core visual reasoning abilities independent of language knowledge, revealing that even top-tier VLMs significantly lag behind 6-year-old children on basic visual tasks. InkSlop focuses on digital-ink spatial reasoning, testing models on maze solving, handwriting trajectory completion, and overlapped handwriting recognition. Together, they provide an intuitive assessment of whether VLMs possess genuinely grounded visual intelligence rather than relying on linguistic shortcuts. A subset of these tasks has been investigated in recent work, as illustrated in Fig. 12 of Kimi-K2.5 (Team et al., 2026).

## 7. Alternative Views

Our position is that current VLMs seldom truly "think" with interleaved images in their CoT, and detailed ablation studies should be reported for transparency. One may argue that the cause of this phenomenon is that the interleaved images are not faithful, e.g., the crops do not contain the target (Hou et al., 2025). However, this occurs only when the faithfulness of the crops does not contribute much to correctness in the context of rule-based reinforcement learning. Therefore, the ablation on the interleaved images in Section 4 is still required even if the faithfulness is addressed. Another argument would be that the benchmark is trivial for VLMs, thus eliminating the necessity of using interleaved images. We agree that harder benchmarks should be proposed in future "Thinking with Images" research. But we also point out that only tasks that require fundamentally different abilities are useful. Combining multiple difficulties that do not need interleaved images will not make an essential change for VLMs. Additionally, we want to clarify that this paper is not intended to undermine previous "Thinking with Images" methods, which we recognize as pioneering and excellent. Instead, our goal is to stimulate discussion on the underlying reasons for their performance gains and the optimal use of visual information.

## 8. Conclusion

In this paper, we revisit the necessity of interleaved images in the "Thinking with Images" paradigm. These images are often generated through tool-calling and code execution, incurring high costs for both training and inference. Our extensive empirical evidence shows that these interleaved images contribute minimally to the success of recent VLMs. We suggest lightweight alternatives that improve the clarity of "Thinking with Images" methods without imposing extra burdens on researchers. By pinpointing the true factors of performance, we hope to inspire future work that better balances effectiveness and efficiency. This shift mirrors the evolution in textual CoT, where rethinking long CoTs led to more adaptive strategies. Finally, we provide recommendations for building next-generation "Thinking with Images" benchmarks and visual tools to advance the field.

## Acknowledgments

We are grateful to the anonymous reviewers for their valuable comments and suggestions that greatly enhanced the clarity of the manuscript. One such improvement, prompted by their feedback, was moving the bonus section to the Appendix.

The computations in this research were performed using the CFFF platform of Fudan University.

This work is supported by National Natural Science Foundation of China No. U2241212, No. 62276066.

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

## A. Bonus: A Simple Way to Achieve Better Performance

*Table 5.* **Results on V\* (Wu & Xie, 2024) combining VLMs with Hide (Liu et al., 2025a).** "Better context" refers to augmenting visual evidence while avoiding additional textual context, and "Better prior" denotes finetuned priors. Best results are **bold**.

| Method | Better context | Better prior | Attribute | Spatial | Overall |
|---|---|---|---|---|---|
| Qwen2.5-VL-7B | ✗ | ✗ | 77.4 | 75.0 | 76.4 |
| Qwen2.5-VL-7B+Hide | ✓ | ✗ | 88.7 | 84.2 | 86.9 |
| DeepEyes-7B | ✗ | ✓ | 84.3 | 84.2 | 84.3 |
| DeepEyes-7B+Hide | ✓ | ✓ | **89.6** | **85.5** | **88.0** |

Based on the discussions in §5, the underlying reasons for the performance gains achieved by "Thinking with Images" methods become clear. Building on these insights, we propose a simple strategy to further improve performance on perception benchmarks such as V\* (Wu & Xie, 2024). We present this method as a bonus, which can be safely skipped by readers who are unfamiliar with visual tools or are not primarily concerned with benchmark performance.

As discussed in Section 2, several recent works have pointed out that VLMs often already "know where to look" on visual tasks (Zhang et al., 2025a; Liu et al., 2025a;b). Their failures on certain examples are largely due to harmful contextual information, such as spurious language priors or irrelevant image backgrounds. A striking example is shown in Figure 3 of Liu et al. (2025b), where VLMs incorrectly infer the brand of a nano receiver as Logitech due to the dominance of a misleading text prior.

These discussions suggest two straightforward directions for improvement. (i) **Better context:** The most useful visual context typically lies in the image regions containing the target object. Cropping these regions and appending them to the input context can therefore be beneficial. However, RL-based "Thinking with Images" methods evaluated in this paper generate such crops through tool-calling or code execution, which inevitably introduces additional textual context alongside the visual crops. When the original query text is harmful, this extra textual context, often generated under the influence of the same harmful prior, is likewise harmful. A safer alternative is to append only the cropped image regions, as done in Vicrop (Zhang et al., 2025a) and Hide (Liu et al., 2025a). (ii) **Better prior:** For certain downstream benchmarks, finetuning VLMs can help mitigate harmful priors. We believe this mechanism largely explains the performance gains observed with RL-based "Thinking with Images" methods.

Notably, these two directions are complementary. To obtain the best of both worlds, we apply Hide (Liu et al., 2025a) to the DeepEyes-7B (Zheng et al., 2026) checkpoint. The results are shown in Table 5. We observe that both improved visual context and improved text priors contribute positively to performance, and that combining them yields further gains. The benefit of simply combining these two methods further highlights the need to rethink the performance gains of "Thinking with Images" methods and the optimal approach to using interleaved images.

