# OpenReview forum: "Position: Your VLM May Not Be Thinking with Interleaved Images"
_ICML.cc/2026/Position_Paper_Track — ICML 2026 Position Paper Track regular_

### Official Review · Reviewer_92tq · 2026-02-28

**Significance:** 3
**Argument Clarity:** 3
**Rating:** 5
**Confidence:** 4

**Questions:**

See the second point in the weaknesses.

**Alternative Views Section:**

Yes

**Compliance With Llm Reviewing Policy A Conservative:**

Affirmed.

**Discussion Potential:**

3

**Final Justification:**

This work proposes a very different point to the other research papers: do VLMs really think with interleaved images? The experiments are simple yet effective to support this proposal. The authors try their best to resolve my concerns during rebuttal. I think it has a great value to the whole community: rethinking our ways to enable VLM thinking/reasoning. Therefore, I recommend a clear accept.

**Paper Summary:**

This work revisits the necessity of interleaved images in a recently hot domain - "thinking with images", where interleaved images are often generated through tool-calling and code execution from the original image and incorporated in the reasoning trajectory for the purpose of improving performance. It demonstrates that interleaved images do not play a significant role in the success of recent "thinking with images" methods. Instead, the improved language generation distribution is the main source of performance gain.

**Position:**

Yes

**Position In Title:**

Yes

**Related Work:**

3

**Strengths And Weaknesses:**

Strengths:
- This work presents a clear position.
- It designs simple yet effective empirical studies to demonstrate the stated position.
- The focused topic is very meaningful and could provide insight to a broader community.

Weaknesses:
- I think Figure 3 is missing. Without it, this work is not complete. This is my main reason for rejection. If I made a mistake, please point that and I would like to raise my score.
- Section 5.2 is not strong enough to me. First, what is the performance of the original version of DeepEyes, Pixel-Reasoner and Thyme (the ones without mased images)? With this result, we could have deeper understanding for the underlying reason for performance gain. Second, which images are masked? both original input image and interleaved images, or simply the original image image or only the interleaved image?
- It would be better if there is more discussion on large-scale VLMs. I understand that this could be difficult due to limited computing resource and the authors try to explain that by referring to community results with Table 2 (which is not strong enough to support the statement proposed in this work). However, LLMs/VLMs with different parameter scales usually have different behaviors. There is a possibility that large-scale VLMs could benefit from interleaved images while small-scale VLMs could not.

**Support:**

3

---

> ### Author Rebuttal · Authors · 2026-03-25
>
> Dear Reviewer 92tq,
>
> Thank you very much for your time and effort in reviewing our paper, especially under the strict Policy A. We are grateful for your thoughtful feedback and for your positive comments on the topic and our design choices. Below, we address your concerns one by one.
>
> > W1: I think Figure 3 is missing. Without it, this work is not complete. This is my main reason for rejection. If I made a mistake, please point that and I would like to raise my score.
>
> Do you mean the sentence in line 337 where we state, "A striking example is shown in Figure 3 of Liu et al. (2025b)"? By this, we are referring to the third figure of [1] (the other paper), not our submission. We could not copy their figure into our submission due to copyright. We thank you for your understanding.
>
> [1] Liu, Zhining, et al. "Seeing but not believing: Probing the disconnect between visual attention and answer correctness in vlms." arXiv preprint arXiv:2510.17771 (2025).
>
> > W2: Section 5.2 is not strong enough to me. First, what is the performance of the original version of DeepEyes, Pixel-Reasoner and Thyme (the ones without mased images)? With this result, we could have deeper understanding for the underlying reason for performance gain. Second, which images are masked? both original input image and interleaved images, or simply the original image image or only the interleaved image?
>
> The performance of the original versions of these models is already reported in Table 1. In Table 4, we did not repeat those numbers in order to avoid redundancy.
>
> Regarding the masking protocol, we mask the target object in the original input image. Since the interleaved images are cropped from the original image, the target object is correspondingly masked in those cropped images as well.
>
> > W3: It would be better if there is more discussion on large-scale VLMs. I understand that this could be difficult due to limited computing resource and the authors try to explain that by referring to community results with Table 2 (which is not strong enough to support the statement proposed in this work). However, LLMs/VLMs with different parameter scales usually have different behaviors. There is a possibility that large-scale VLMs could benefit from interleaved images while small-scale VLMs could not.
>
> Thank you very much for the suggestion. We agree that large-scale VLMs are very important. We now have the following reproduced results, which were conducted by ourselves last month:
>
> |                             | HR-Bench4K | HR-Bench8K | Overall |
> |-----------------------------|------------|------------|---------|
> | Qwen3-VL-235B-A22B-Instruct | 85.6       | 82.3       | 84.0    |
> | +Thinking with Images       | 85.4       | 82.4       | 83.9    |
>
> This result aligns with our findings on the large-scale VLMs and also aligns with community results. We will incorporate this additional evidence and discussion to strengthen the paper.
>
> Thank you again for your constructive comments and encouragement. We sincerely appreciate your consideration and hope our clarification helps address your concerns.
>
> Best regards,
>
> Authors

---

> > ### Author Rebuttal · Reviewer_92tq · 2026-04-03
> >
> > Thanks for your response. I don't have further questions.

---

### Official Review · Reviewer_b1DP · 2026-03-01

**Significance:** 4
**Argument Clarity:** 3
**Rating:** 5
**Confidence:** 4

**Questions:**

In the masked experiment, models still outperformed the base Qwen2.5-VL-7B even when targets were occluded. Do you suspect this is strictly due to better language priors, or could the model be relying on contextual visual correlations? Perhaps one experiment to further understand it is to remove the images entirely or giving an empty image and check the QA performance.

**Alternative Views Section:**

Yes

**Compliance With Llm Reviewing Policy A Conservative:**

Affirmed.

**Discussion Potential:**

3

**Paper Summary:**

The paper posits that recent Vision-Language Models (VLMs) operating under the "Thinking with Images" paradigm do not meaningfully rely on the interleaved images generated during their Chain-of-Thought (CoT) reasoning. Through empirical analysis, including ablation studies and attention visualizations, the authors argue that the observed performance gains stem primarily from the improved language generation distribution resulting from fine-tuning, rather than from the active utilization of interleaved visual information.

**Position:**

Yes

**Position In Title:**

Yes

**Related Work:**

3

**Strengths And Weaknesses:**

## Strengths
- The authors cleanly identify a timely issue for the popular "think with images" and systematically dismantle the assumption that interleaved images are the core driver of performance. They effectively use ablation studies (inference-time image ablation, vanilla SFT control, and targeted visual masking) to prove this. Especially, by suppressing tool-generated images while retaining the textual reasoning, the resulted negligible performance degradation across multiple benchmarks supports the position.
- The paper is well-written, easy to follow, and employs interesting attention rollout visualizations. These visualizations provide qualitative evidence that the models predominantly attend to the original input images rather than the newly generated interleaved images during reasoning.


## Weakness
1. The core empirical evaluations (DeepEyes, Pixel-Reasoner, Thyme) are all models fine-tuned from the exact same base architecture, Qwen2.5-VL-7B. While this might be a limitation from the model developer, the conclusion from the ablation studies can only be certain for Qwen model but unsure if it can generalize to other models (however, the proposal of adding an ablation study can be applied in future work).
2. The paper acknowledges that current methods primarily use cropping to mimic zoom-in behaviors. As the paper essentially proves that cropping is not the main reason for the improvement of "think with images", it slightly overclaims by framing this as interleaved images broadly.

**Support:**

4

---

> ### Author Rebuttal · Authors · 2026-03-25
>
> Dear Reviewer b1DP,
>
> Thank you very much for your time and effort in reviewing our paper, especially under the strict Policy A. We are truly encouraged by your supportive comments and sincerely appreciate your recognition of our work. Below, we address the concerns you raised point by point.
>
> > W1: The core empirical evaluations (DeepEyes, Pixel-Reasoner, Thyme) are all models fine-tuned from the exact same base architecture, Qwen2.5-VL-7B. While this might be a limitation from the model developer, the conclusion from the ablation studies can only be certain for Qwen model but unsure if it can generalize to other models (however, the proposal of adding an ablation study can be applied in future work).
>
> Thank you for this insightful suggestion. We agree that the current "Thinking with Images" models are predominantly built upon Qwen2.5-VL-7B. We believe this is largely due to the strong open-source foundation provided by the Qwen2.5-VL series. To further examine the generalizability of our findings, we conducted additional experiments over the past month on Qwen3-VL, which natively supports the "Thinking with Images" feature through the official release. The results are shown below:
>
> |                             | HR-Bench4K | HR-Bench8K | Overall |
> |-----------------------------|------------|------------|---------|
> | Qwen3-VL-235B-A22B-Instruct | 85.6       | 82.3       | 84.0    |
> | +Thinking with Images       | 85.4       | 82.4       | 83.9    |
>
> These results are consistent with our main findings and also align with the community results discussed in the paper. As you precisely pointed out, comparable "Thinking with Images" models are still limited in other VLM families. Nevertheless, we believe our ablation framework can be readily applied to future models once such systems become available.
>
> > W2: The paper acknowledges that current methods primarily use cropping to mimic zoom-in behaviors. As the paper essentially proves that cropping is not the main reason for the improvement of "think with images", it slightly overclaims by framing this as interleaved images broadly.
>
> Thank you very much for this helpful comment. When writing our paper, we especially try to avoid overclaiming (this is the reason for the "may not" in the title). We will revise the abstract and introduction to more clearly describe the scope of our claim, especially with respect to the role of cropping-based interleaved images. We believe this issue can be addressed through clearer wording and does not affect the core findings of the paper.
>
> > Q1: In the masked experiment, models still outperformed the base Qwen2.5-VL-7B even when targets were occluded. Do you suspect this is strictly due to better language priors, or could the model be relying on contextual visual correlations? Perhaps one experiment to further understand it is to remove the images entirely or giving an empty image and check the QA performance.
>
> Thank you very much for the suggestions. We believe the language generated distribution $P(\mathrm{language}\\, \mathrm{output}|\mathrm{language}\\, \mathrm{prompt}, \mathrm{visual}\\, \mathrm{background})$ plays a vital role, which can be viewed as a combination of language priors and contextual visual correlations. We will modify the paper to clarify that. For the experiment you suggested, we will conduct it as soon as possible and report to you once complete.
>
> Thank you again for your support and constructive feedback. We sincerely appreciate it and wish you the best of luck with your own submission.
>
> Best regards,
>
> Authors

---

> > ### Author Rebuttal · Reviewer_b1DP · 2026-04-04
> >
> > The authors have addressed my concerns. I will keep my original score.

---

### Official Review · Reviewer_F1Pk · 2026-03-08

**Significance:** 4
**Argument Clarity:** 2
**Rating:** 3
**Confidence:** 5

**Questions:**

1. The ablation studies presented in the paper are carried out exclusively on VQA-style benchmarks that may not, by their very design, require any meaningful visual interaction. Would it be possible to extend the evaluation to tasks where the functional necessity of interleaved images is less ambiguous, geometric reasoning or maze-solving, for instance, as the authors themselves gesture toward in Section 6, and to report how ablation performance on such tasks compares against the existing benchmark results? In the absence of this kind of evidence, the paper's central argument is at risk of circularity: the conclusion that interleaved images are unhelpful is drawn from benchmarks that arguably never required them in the first place.

2. Regarding Table 3: the SFT baseline is trained on data that was originally constructed for the DeepEyes RL pipeline, which may already be closely aligned in distribution with the evaluation benchmarks being used. How do the authors rule out the possibility that the observed performance parity between SFT and "Thinking with Images" training simply reflects an overlap in data distribution, rather than a more fundamental equivalence between the two training regimes? Without controlling for this confound, the comparison is difficult to interpret with confidence.

**Alternative Views Section:**

Yes

**Compliance With Llm Reviewing Policy A Conservative:**

Affirmed.

**Discussion Potential:**

4

**Final Justification:**

I increased the score to 3.

**Paper Summary:**

"Thinking with Images" has recently gained considerable traction as an emerging research direction within the Vision–Language Model (VLM) community. The underlying idea is that, at inference time, a model may call upon external tools or run code to produce intermediate visual outputs—such as cropped regions or rescaled images—which are then woven into the chain-of-thought reasoning process to assist in solving visually grounded tasks.

Despite the appeal of this paradigm, the authors challenge the assumption that existing VLMs are actually leveraging these injected visuals in any meaningful way during reasoning. They contend that the performance improvements typically attributed to visual interleaving arise from other sources entirely. Drawing on ablation experiments targeting the interleaved images and on attention pattern analyses conducted via Attention Rollout, they show that the dominant driver of improvement is fine-tuning itself—which shifts the model's language generation distribution closer to the target benchmarks—rather than any genuine informational contribution from the visuals being inserted.

The paper goes on to suggest that the "Thinking with Images" research agenda may be afflicted by a deeper, systemic misreading of results: significant computational investment is being channeled into sustaining image-interleaving infrastructure, while the actual downstream benefit this mechanism provides proves to be minimal. On these grounds, the authors call for the community to step back and critically reexamine whether this line of inquiry is delivering on its promises.

**Position:**

Yes

**Position In Title:**

Yes

**Related Work:**

3

**Strengths And Weaknesses:**

**Strengths:**

1. The research question is well-timed and fits naturally within the scope of ICML. The paper's core provocation—that interleaved visual inputs in VLMs may contribute far less to the reasoning process than commonly assumed—is a genuinely interesting claim, especially in light of the substantial computational costs that accompany the integration of interleaved visual and textual content.

2. To build its case, the paper draws on a fairly broad methodological toolkit, incorporating ablation experiments on interleaved images, Attention Rollout-based attention analysis, supervised fine-tuning investigations, and evaluations that employ input masking.

**Weaknesses:**

1. The paper is undermined by a significant internal inconsistency. The authors concede in Section 7 that the benchmarks they rely on may not be well-matched to the phenomenon under investigation. The chosen benchmarks—V*, HR-Bench, ChartQA, and OCRBench—are designed around VQA settings that call for direct, single-pass inference over a static input image, with no CoT reasoning involved. Deploying these benchmarks to interrogate the functional role of interleaved image tokens in a CoT-based reasoning pipeline is an inherently skewed test, since the tasks themselves never demanded the type of iterative, multi-step visual reasoning that image interleaving is supposed to enable. The situation is not unlike trying to determine whether a specialized instrument is necessary by applying it to a context where it was never intended to be used. As a result, the paper's headline conclusion—that VLMs do not genuinely reason with interleaved images—is not adequately grounded in the evidence the authors present.

2. A number of methodological issues warrant more careful treatment. The heavy reliance on Attention Rollout is a notable vulnerability: the method is an approximation whose capacity to accurately characterize information flow becomes increasingly unreliable under the complex dynamics of multi-head attention and residual connections. A substantial line of prior work—including the GradCAM family of attribution methods—has catalogued the limited correspondence between raw attention weights and actual model behavior, yet the authors largely sidestep these documented concerns. The qualitative evidence assembled in Figure 2 is also too thin to support the interpretive load placed on it: three cherry-picked examples, one per model, fall well below any threshold for statistical credibility. Since the authors themselves treat the ablation studies as the more dependable strand of evidence, the attention visualizations contribute relatively little and carry a real risk of misleading readers.

Beyond these issues, the interpretation drawn from Table 4 involves an inferential leap that the data cannot fully sustain. The finding that masked models continue to outperform baselines does not, on its own, isolate language prior alignment as the exclusive source of improvement. An equally coherent reading is that fine-tuning simultaneously sharpens both language priors and visual perception, with masking simply making the former more visible. These two mechanisms are not mutually exclusive, and the experimental setup as designed lacks the resolution to pull them apart.

**Support:**

3

---

> ### Author Rebuttal · Authors · 2026-03-25
>
> Dear Reviewer F1Pk,
>
> Thank you very much for your time and effort in reading our paper, especially under the strict policy A. We thank you for your appreciation of the timeliness and the topic of our paper. Below, we discuss your concerns one by one.
>
> > W1 & Q1: The chosen benchmarks—V*, HR-Bench, ChartQA, and OCRBench—are designed around VQA settings...Would it be possible to extend the evaluation to tasks where the functional necessity of interleaved images is less ambiguous, geometric reasoning or maze-solving, for instance, as the authors themselves gesture toward in Section 6...
>
> Thank you for your question. We are glad to see that we agree future benchmarks should focus more on geometric reasoning or maze solving, where Visual CoT is truly needed. However, we are writing a position paper which, per the ICML 2026 Position Papers guidelines, makes an argument for a viewpoint or perspective about what *should* be done. As our title, "Your VLM May Not Be Thinking with Interleaved Images," suggests, we are arguing that interleaved images do not play a vital role in *current* "Thinking with Images" methods. We never argue that thinking with interleaved images will not work in the future. In fact, we believe this is a highly promising research direction, which is exactly why we wrote this position paper.
>
> Regarding the evaluation, these VQA benchmarks are not selected by us, they are studied by the "Thinking with Images" community as the standard for almost a year.  Part of the motivation for this position paper is to call for a shift away from these VQA benchmarks toward the geometric or maze-solving tasks you mentioned. We are aware of some excellent benchmarks released just around the submission deadline (Jan. 28), e.g., BabyVision (Jan. 12) and InkSlop (Jan. 27), and we will add them into our Call For Action section. Although we are actively working on building these new benchmarks in our other projects, we believe accomplishing them is beyond the scope of a position paper. With this clarification, we hope you will see that we hold the exact same position.
>
> > W2: A number of methodological issues warrant more careful treatment. The heavy reliance on Attention Rollout is...
>
> We thank you for your cautions. However, we must respectfully defend our work against the suggestion that it might mislead readers, as this is something we would never do. In fact, in Line 264, we explicitly state that "while attention visualizations provide qualitative insights, we consider the quantitative results from the ablation study to be more robust evidence of the model behavior." Furthermore, we are disheartened to see our examples in Figure 2 characterized as "cherry-picked." For instance, the DeepEyes example shown is simply the very first instance in the V* benchmark, not a curated selection. Since we rely on the ablation studies for our core evidence, the visualizations are included purely as supplementary qualitative insights.
>
> > W3: Beyond these issues, the interpretation drawn from Table 4...
>
> Thank you for this suggestion. We apologize for any misunderstanding, but we are not trying to say that language priors are the only factor in the improvement. We are trying to say, as in line 372, that a better language generation distribution, i.e., $P(\mathrm{language}\\, \mathrm{output}|\mathrm{language}\\, \mathrm{prompt}, \mathrm{visual}\\, \mathrm{background})$ is the primary source. We will modify the wording in our revision to make this distinction clearer.
>
> > Q2: Regarding Table 3: the SFT baseline is trained on data that was originally constructed for the DeepEyes RL pipeline, which may already be closely aligned in distribution with the evaluation benchmarks being used. How do the authors rule out the possibility that the observed performance parity between SFT and "Thinking with Images" training simply reflects an overlap in data distribution, rather than a more fundamental equivalence between the two training regimes? Without controlling for this confound, the comparison is difficult to interpret with confidence.
>
> Thank you for your question. As mentioned earlier, our objective is not to prove that SFT and "Thinking with Images" training are fundamentally equivalent, nor are we trying to claim that "Thinking with Images" will never work. We are simply arguing that current VLMs may not be effectively "Thinking with Images" at this present stage.
>
> Thank you again for your detailed review. We believe that after these clarifications, it is evident that we actually share very similar ideas. We would be happy to discuss further how to push the community to study more interesting tasks that generally require Visual CoTs!
>
> Best regards,
>
> Authors

---

> > ### Author Rebuttal · Reviewer_F1Pk · 2026-04-02
> >
> > The reviewer appreciates the author's feedback. However, I still have concerns about the paper's motivation. Therefore, I only increased the score to 3.

---

### Official Review · Reviewer_FikN · 2026-03-13

**Significance:** 2
**Argument Clarity:** 3
**Rating:** 4
**Confidence:** 4

**Questions:**

- If the proposed solutions are adopted and training focuses more on textual reasoning and direct improvement through supervision, can we expect VLMs to learn about active perception and autonomous information retrieval? How would this capability be induced?
- Assuming that the benchmarks can be largely solved without requiring "thinking with images", what is the implication on the experimental results? While it remains true that the models do not make use of cropped images, the training algorithms/data might not be invalidated.
- What exactly is meant by "language generation distribution" in Takeaway 3? Is it aligning the language with the task? Or does it also imply a shift in its reasoning behavior tailored to images?

**Alternative Views Section:**

Yes

**Compliance With Llm Reviewing Policy A Conservative:**

Affirmed.

**Discussion Potential:**

2

**Final Justification:**

The authors were able to address my concerns mostly revolving around the mismatch between the call for actions and the initial premise and experiments around "thinking with interleaved images".

The authors provided a comprehensive plan to update the paper which better puts the experiments around SFT and alternatives to thinking with images into perspective of incentivizing proper benchmarking and goal-oriented approach towards visual reasoning.

As a result I am increasing my score to 4. However, this score is contingent to all the promised updates to the paper from the rebuttal.
One reservation I have is that, since I don't know how the final revision will turn out, there is a risk in accepting the paper due to the large amount of changes required.

**Paper Summary:**

The paper argues that current VLMs that have been trained to "think with images", i.e., use chain of thought (CoT) to solve complex vision tasks, seldom utilize the visual images inside the thinking process effectively. It is shown that VLMs perform almost equally when the image is removed from the reasoning process and the language maintained. Hence, the performance improvement of these VLMs stems from better language capabilities and alignment with the task rather than obtaining new visual information through tool calling. The paper advocates for a more rigorous evaluation of CoT VLMs and for the development for better tooling and benchmarks to truely evaluate "thinking with images".

**Position:**

Yes

**Position In Title:**

Yes

**Related Work:**

3

**Strengths And Weaknesses:**

Strengths:
- The position is well motivated, and presented.
- The ablation study provides strong support for the main argument of this position: VLMs do not rely on the interleaved image for the performance improvement
- Other experiments provide evidence for the additional argumentation that improve language (reasoning) on images is likely the main contributor
- The position is likely to resonate with most of the ICML community as the results are clear.

Weaknesses:
The main concern of this position paper is that it adopt a narrow view of the field at this very moment without projecting toward more long-term goals:
- In section 5.3, the paper discusses directions for improvements, mainly "better context" and "better text prior". These directions somewhat imply that research should focus less on performing reasoning optimization (with RL) and adopt more direct approaches. However, this view fundamentally ignores the bigger picture. Yes, these suggestions may improve on current benchmarks, but many complex real-world scenarios require active perception where agents seek out information even outside the scope of the original observation. Here, but also the paper as a whole focuses entirely on image-level reasoning and cropping as the main tool, although this research likely paves the way for more complex tasks. Discouraging performing visual reasoning during training through complex and expensive learning mechanisms (e.g. RL) could slow the progress that is needed to eventually tackle much more challenging tasks.
- The most interesting part of the paper are the call for actions that clearly identify that we require tasks with tool calling going beyond cropping and with benchmarks that necessitate visual reasoning and acquiring of novel information. Overall, I find this narrative to be underdiscussed and underexplored as the position paper could provide a more concrete and opinionated way forward.

The paper adopt a strong model-view as opposed to discussing the relevance of the benchmarks more:
- Experimental support largely treats benchmarks (and to some extend the training data) as ground truth, not questioning enough their suitability for evaluating visual CoT. Instead, the position focuses on the behavior of the models. While this view is certainly interesting, a more complete position could have taken both variables into account. If the benchmarks are not measuring "thinking with images" it might not be a good way to argue these models cannot visually reason.
- Despite the strong focus on models, it is not entirely clear what exactly drives the performance improvements of the visual reasoning models. The reason could be a generic alignment with the task solely obtained through fine-tuning or a more fundamental shift in the model's reasoning behavior that enriched its ability to reason about visual concepts and contrasting image regions.

Overall, while the position is well argued and easy to approved of, it seems to be too much of a snapshot of issues with current models, data, and benchmarks. As the field moves forward, the position could quickly lose relevance as we approach more challenging tasks that go beyond image-level reasoning. Therefore, I am leaning towards "weak reject".

**Support:**

3

---

> ### Author Rebuttal · Authors · 2026-03-25
>
> Dear Reviewer FikN,
>
> Thank you very much for your time and thoughtful review, especially under Policy A. We sincerely appreciate your positive feedback on our motivation and presentation. Below we address your concerns point by point.
>
> > W1: In section 5.3, the paper discusses directions for improvements...The most interesting part...
>
> Thank you for raising this. Our intention is not to discourage visual reasoning training, including reinforcement learning. Rather, the experiment in Sec. 5.3 is intended as a proof-of-concept study of how the Crop tool can be used better in the current setting, which is why we explicitly label this section as a "bonus".
>
> More broadly, the message of our paper is that the current combination of narrow tools and tasks may not be sufficient for advancing toward true visual intelligence. We fully agree that real-world settings will require active perception and information beyond cropping. In fact, this is precisely why our Call to Action advocates for tasks that genuinely require reasoning with images and more diverse forms of interaction. We also agree that such capabilities may ultimately be developed through methods such as RL.
>
> > W2: Experimental support...
>
> We agree that benchmark suitability is an important part of the discussion. The benchmarks studied in our paper were not chosen arbitrarily by us, they have served as standard evaluation settings in the recent "Thinking with Images" literature for nearly a year. One of the motivations of our position paper is precisely to encourage the community to move beyond these benchmarks toward tasks that more directly require visual reasoning.
>
> We are also aware of several promising new benchmarks released close to the submission deadline (e.g., BabyVision, Jan. 12, and InkSlop, Jan. 27), and we will mention them in the Call to Action section. At the same time, we believe that a full empirical study of current "Thinking with Images" methods on these newly introduced benchmarks would be beyond the scope of a single position paper, especially given the very recent timing of their release.
>
> > W3: Despite the strong...
>
> Thank you for the question. Identifying the source of the performance gain is in fact one of the main goals of our empirical analysis. Section 5 is explicitly titled “The underlying reason for the performance gain,” where we show that vanilla SFT can largely match the performance of "Thinking with Images" models.
>
> > W4: Overall, while the position is well argued... the position could quickly lose relevance...
>
> We truly appreciate your recognition. However, we respectfully disagree this criticism. The concern that the position could “quickly lose relevance” as the field moves toward more challenging tasks seems to us to describe the expected role of a timely position paper, rather than a weakness.
>
> As the ICML guidelines state, a position paper is meant to "make an argument for a viewpoint or perspective about what should be done." Once the actions advocated by such a paper are taken, it is natural that the paper may become less central to the field. In that sense, if our paper becomes less relevant because the community moves beyond the current narrow setting, that would suggest the paper has served its intended purpose.
>
> We do not think a position paper should be expected to remain perpetually relevant by addressing a problem setting that can never be resolved. We nevertheless respect your concern and welcome further discussion.
>
> > Q1: If the proposed solutions...
>
> Thank you for the question. We would like to clarify that we are not advocating abandoning "Thinking with Images" methods in favor of purely textual reasoning. Our point is instead that the current tools and benchmarks in this line of work are too narrow, and should be expanded to include more diverse forms of interaction and more challenging tasks. We view "Thinking with Images" as a promising direction, which is precisely why we wrote this paper.
>
> > Q2: Assuming that the benchmarks...
>
> As mentioned earlier, the implication of this experimental result is that we should use more diverse tools and solve benchmarks that truly require visual reasoning. These calls made by us will help guide the community into a better direction for "Thinking with Images". Furthermore, we agree with you that the training algorithms and data are very useful. Additionally, we have explicitly expressed our gratitude to previous "Thinking with Images" methods in the related work section, where we used words like "impressive" and "novel" to describe them. We by no means underestimate these works.
>
> > Q3: What exactly is meant by "language generation distribution"...
>
> By language generation distribution, we mean a better $P(\mathrm{language}\\, \mathrm{output}|\mathrm{language}\\, \mathrm{prompt}, \mathrm{visual}\\, \mathrm{background})$ for the task.
>
> Thank you again for your detailed review. We welcome any further discussion and wish you the best of luck with your own submission.

---

> > ### Author Rebuttal · Reviewer_FikN · 2026-04-02
> >
> > I thank the authors for their clarifications that indicate that their views largely overlap with the criticism of my review.
> >
> > Although Sec. 5.3 is marked as "bonus", by "proposing a simple strategy to further improve performance on perception benchmarks", the paper's narrative is influenced towards solving existing benchmarks with alternatives to "thinking with images". In my opinion, this hurts the bigger picture.
> > This similarly applies to the focus on SFT as an alternative solution.
> >
> > > Thank you for the question. Identifying the source of the performance gain is in fact one of the main goals of our empirical analysis. Section 5 is explicitly titled “The underlying reason for the performance gain,” where we show that vanilla SFT can largely match the performance of "Thinking with Images" models.
> >
> > This does not answer the question if SFT leads to a generic alignment with the task or a more fundamental shift in the model's reasoning behavior that enriched its ability to reason about visual concepts and contrasting image regions.
> >
> > Overall, the in the current paper version, the authors' strong opinion with the call for actions is not clear to the reader throughout the paper, which, in my opinion, hurts the paper's impact since it is perceived to be much narrower (only discussing current "thinking with images" approaches).
> >
> > It would help if the authors could sketch the planned changes to the paper that would resolve the alignment issues between the call for actions and the main paper's arguments towards "thinking with images" as indicated by the initial review and as discussed in the rebuttal.

---

### Decision · Program_Chairs · 2026-04-30

**Decision:**

Accept (regular)

**Comment:**

This position paper argues that Vision-Language Models (VLMs) do not genuinely rely on interleaved visual inputs during Chain-of-Thought reasoning. Through ablation studies, the authors demonstrate that performance improvements stem primarily from improved language generation distributions resulting from fine-tuning, rather than active visual utilization.
Reviewers b1DP and 92tq commended the paper for identifying a timely issue and designing effective empirical studies. Initial concerns highlighted a narrow focus on existing VQA benchmarks that may not require multi-step visual reasoning (Reviewers F1Pk, FikN), and a reliance on a single base architecture (Reviewer b1DP). The authors constructively addressed these by providing additional evaluations on the larger Qwen3-VL model and restructuring the narrative to advocate for more advanced visual reasoning benchmarks.
The paper offers a well-supported, critical perspective that will push the community to rethink multimodal evaluations.